# The Impact of COVID-19 and Related Public Health Measures on Hepatitis C Testing in Ontario, Canada

**DOI:** 10.3390/v17091163

**Published:** 2025-08-26

**Authors:** Yeva Sahakyan, Samantha S. M. Drover, Zoë R. Greenwald, William W. L. Wong, Alexander Kopp, Richard L. Morrow, Naveed Z. Janjua, Beate Sander

**Affiliations:** 1Health Systems and Policy Research Collaborative Centre, University Health Network, Toronto, ON M5G 2C4, Canada; yeva.sahakyan@uhn.ca; 2ICES, Toronto, ON M4N 3M5, Canada; samantha.drover@ices.on.ca (S.S.M.D.); zoe.greenwald@mail.utoronto.ca (Z.R.G.); alexander.kopp@ices.on.ca (A.K.); 3Dalla Lana School of Public Health, University of Toronto, Toronto, ON M5T 3M7, Canada; 4School of Pharmacy, University of Waterloo, Kitchener, ON N2G 1C5, Canada; william.wl.wong@uwaterloo.ca; 5British Columbia Centre for Disease Control (BCCDC), Vancouver, BC V5Z 4R4, Canada; richard.morrow@bccdc.ca (R.L.M.); naveed.janjua@bccdc.ca (N.Z.J.); 6School of Population and Public Health, University of British Columbia, Vancouver, BC V6T 1Z3, Canada; 7Institute of Health Policy, Management, and Evaluation, University of Toronto, Toronto, ON M5T 3M6, Canada; 8Public Health Ontario, Toronto, ON M5G 1M1, Canada

**Keywords:** hepatitis C, COVID-19, population-based cohort, administrative data, priority populations, interrupted time series

## Abstract

The COVID-19 pandemic disrupted progress towards global HCV elimination goals by interrupting essential health services in Canada and globally. We aimed to evaluate the effect of the pandemic on hepatitis C virus (HCV) testing rates in a population-based cohort study in Ontario using health administrative data. All residents with records of either HCV antibody or ribonucleic acid (RNA) tests were included. Monthly testing rate per 1000 population were compared during the pre-pandemic (01/01/2015–29/02/2020) and pandemic (01/03/2020–31/12/2022) periods using interrupted time series models, stratified by sex, homelessness, human immunodeficiency virus (HIV), and immigration status, and people who inject drugs (PWID). The HCV testing rate followed a statistically significant upward trend before the pandemic, dropping at its onset with 1.38/1000 fewer individuals initiating testing monthly. Compared to counterfactual estimates, the observed monthly number of people tested per 1000 population was lower by 1.41 (95% CI: 1.18–1.64) in 2020 (May–Dec), 1.17 (95% CI: 0.99–1.36) in 2021, and 1.41 (95% CI: 1.22–1.59) in 2022, corresponding to relative reductions of 47%, 34%, and 41%, respectively. Testing rates remained below expected levels across all subgroups throughout 2020–2022, with the greatest absolute declines observed among people co-infected with HIV, people experiencing homelessness, and PWID. Tailored, equity-focused interventions are needed to address these persistent gaps in HCV testing, without which Canada’s progress toward its 2030 elimination targets remains at risk.

## 1. Introduction

The COVID-19 pandemic and related public health policies led to widespread disruption in healthcare services in Canada and globally. In Ontario, a state of emergency was declared on 17 March 2020, initiating policies aimed to reduce virus transmission. Public health measures included the suspension of non-essential health services, which affected access to routine and specialized care [1]. Ontario implemented a stay-at-home order as part of its public health response, which likely contributed to reduced access to healthcare services, including HCV testing [2,3]. Laboratory capacity was reallocated to support COVID-19 diagnostics [4], while many community programs, essential for reaching vulnerable populations, were disrupted [5]. Additionally, concerns about infection and mortality risk in medical settings altered healthcare-seeking behavior [6]. The resumption of healthcare services occurred in phases, but pandemic-related policies are likely to continue affecting access to and continuity of care beyond the pandemic period [7,8].

Hepatitis C virus (HCV) remains a major public health concern in Canada, with an estimated 204,000 people living with chronic hepatitis C by the end of 2019 [9]. Ontario has adopted a risk-based screening approach, and reimburses treatment for eligible individuals [10,11]. While elimination efforts were underway prior to the pandemic, significant gaps in HCV testing and care engagement persisted, especially among priority populations [12,13]. The pandemic disrupted HCV care services in Canada, including prevention, screening, diagnosis, and treatment. These disruptions were unlikely to be uniform across the care cascade. The impact on initial screening may differ from that on confirmatory or follow-up testing, a distinction not fully explored beyond the initial waves of the pandemic.

Studies from the provinces of British Columbia and Quebec have highlighted the negative impact of the COVID-19 pandemic on the HCV care cascade, demonstrating reduced testing rates and treatment initiation, particularly for marginalized populations disproportionately affected by HCV [14,15,16]. The impact was short-term with studies showing either full [14] or near-full recovery [15] during the post-pandemic period. In Ontario, the sustained impact beyond the initial waves of the pandemic and related public health measures has yet to be determined. Previous studies demonstrated a 19–35% and 30–44% reduction in HCV antibody and RNA testing rates, respectively [17], and close to 50% reduction in HCV treatment dispensations [18] during the first year of pandemic, with no full recovery during the post-pandemic period.

Pandemic-related disruptions in healthcare services may have long-term consequences for vulnerable populations, including people who inject drugs (PWID), people living with human immunodeficiency virus (HIV), and those experiencing homelessness, who already face barriers in accessing necessary care and are at increased risk of HCV infection, complications, and death [19,20,21]. National guidelines also identify those born between 1945 and 1975 as a high-risk group for whom screening is recommended [22]. Given Canada’s pre-pandemic efforts to eliminate HCV, understanding the pandemic’s impact on the HCV care cascade can help identify gaps in access and continuity of care, determine the most affected populations, and guide resource allocation to advance Canada’s eradication efforts. We hypothesize that the COVID-19 pandemic caused a substantial and sustained reduction in HCV testing rates, with the greatest declines and slowest recovery observed among priority populations. This study aims to evaluate impact of the COVID-19 pandemic and related control measures on hepatitis C testing overall and among priority populations in Ontario, Canada.

## 2. Materials and Methods

### 2.1. Study Design, Setting, and Population

We report our study in accordance with the RECORD statement for observational research [23]. We conducted a population-based cohort study using an interrupted time series approach to estimate the impact of the COVID-19 pandemic on HCV testing rates using health administrative databases. We included all Ontario residents with any record of either HCV antibody or HCV ribonucleic acid (RNA) testing available in electronic health records during pre-pandemic (1 January 2015 to February 2020) and pandemic periods (March 2020–31 December 2022), which included multiple phases of restrictions and re-openings. Each participant’s index date was defined as the earliest available HCV antibody or RNA or genotype test date or antiviral treatment initiation date during the corresponding study windows, to account for variations in test capture across datasets, e.g., where a genotype test date was documented earlier than an antibody or RNA test dates. Only individuals with an HCV antibody or RNA test recorded during the study window were included, and the index date was used solely to anchor baseline characteristics. We excluded individuals without a valid Ontario Health Insurance Plan (OHIP) number, those older than 100 years, individuals with missing age or sex data, and those whose first test date was recorded on or after their date of death.

### 2.2. Variable Definitions and Data Sources

We linked health administrative datasets held at ICES (Appendix A) using unique encoded identifiers and analyzed the data at ICES. ICES is a prescribed entity under Ontario’s Personal Health Information Protection Act (PHIPA). Section 45 of PHIPA authorizes ICES to collect personal health information, without consent. We obtained information on HCV antibody and RNA tests from the Ontario Laboratories Information System (OLIS), a population-wide repository that includes test orders from commercial, hospital, and public health laboratories. We analyzed OLIS data from 2015 onwards, to cover years when Public Health Ontario, the provincial laboratory conducting confirmatory HCV testing, was contributing data to OLIS [24,25]. We considered the following priority populations with elevated risks of HCV infection: people experiencing homelessness, immigrants and newcomers, people living with HIV, and PWID [9,12,19,26,27]. Immigration status was retrieved from the Immigration, Refugees and Citizenship Canada (IRCC) Permanent Resident Database, which records individuals who have landed in Ontario and received Canadian permanent resident status since 1985. Homelessness was defined as a binary variable using a validated algorithm based on diagnostic and postal codes (Appendix A) considering a two-year lookback period [28,29]. HIV was defined using the ICES HIV database, where a person was considered HIV-positive if they had three physician claims with an HIV diagnosis over three years [30]. PWID status was defined using a validated algorithm to detect a history of injecting (at least one physician visit, emergency department visit, or hospitalization for drug dependence, or opioid agonist treatment record) versus no indication of injecting/drug use (Appendix A) [31].

We obtained demographic information from the Registered Persons Database (RPDB). Neighborhood-level dimensions of social marginalization were estimated using the Ontario Marginalization Index Database (ONMARG) and included households and dwellings (previously residential instability), material resources (previously material deprivation), and racialized and newcomer populations (previously ethnic concentration). We used validated algorithms within the administrative data sets to define advanced liver disease including cirrhosis, decompensated cirrhosis, hepatocellular carcinoma, and liver transplantation (Appendix A) [32], and the Johns Hopkins Adjusted Clinical Group^®^ (ACG) [33] system to determine the total number of comorbidities (Aggregated Diagnosis Groups (ADGs) comorbidity classification scheme) over the 2-year period preceding the index date.

### 2.3. Statistical Analysis

Baseline characteristics were summarized at index date using descriptive statistics. The primary outcome was defined as the monthly rate of individuals tested for HCV (AB or RNA testing) calculated per 1000 individuals of the Ontario population. If individuals were tested multiple times within a month, they were counted once. If they were tested in different months, they were counted once for each month. Secondary outcomes were the monthly rates of individuals with AB tests and those with RNA tests, measured separately.

An outlier analysis showed that the number of individuals tested in the months of transitioning from the pre-pandemic to the pandemic period deviated markedly from the overall temporal pattern (Appendix A). We excluded these transition months (February, March, and April of 2020) to obtain trends that were closer to linear and to improve model interpretability. While non-linear models might better capture the initial drop in testing volume, we chose linear models to focus on sustained changes in testing patterns rather than the acute, short-term impact of the pandemic.

To estimate the impact of the pandemic on testing rates we first fitted the interrupted time series regression model, which included a binary variable for the study period (pre-pandemic vs. pandemic) and two time variables (in months) for assessing trends during pre-pandemic and pandemic periods. We observed statistically significant autocorrelation and the models were adjusted accordingly. Seasonality, evaluated using Yule–Walker estimates, had no significant effect and, thus, was excluded from the adjusted model.

Next, we used pre-pandemic estimates from the model to generate counterfactual values, which represent the expected number of individuals tested had the pre-pandemic trends continued uninterrupted into the pandemic phase. These counterfactual estimates serve as a reference to quantify the impact of the pandemic by comparing observed testing rates to projected (i.e., counterfactual) values under a no-pandemic scenario. Finally, to estimate the difference between observed and counterfactual testing rates during pandemic, we fitted a second model to a dataset containing both actual and counterfactual testing rates. This model included a binary variable for observed vs. counterfactual rates, a time variable representing months since start of the study (1 January 2015), the years 2021 and 2022 (2020 as the reference year), and two interaction terms between the years and the binary variable. The standard errors for the observed–counterfactual differences were computed based on the Yule–Walker covariance matrix.

To evaluate HCV testing rates among priority populations we stratified the analysis by sex, birth cohort, immigration status, homelessness, HIV, and PWID status. To obtain stratified denominators, we assigned a pseudo-index date to all individuals recorded in the RBDB database. We then randomly sampled 20% of this population, restricting the sample to individuals who were <100 years old, OHIP-eligible, and alive on the pseudo-index date. Covariate values were computed as of the pseudo-index date. All analyses were conducted at ICES using SAS statistical software (SAS Enterprise Guide 8.3).

## 3. Results

### 3.1. Characteristics of the Study Cohort

In total, we identified 3,202,116 individuals with any HCV antibody or RNA or genotype test or treatment initiation or sustained virologic response (SVR) attainment during the study timeframe. Among them 3,193,543 met eligibility criteria (Appendix A).

During the pre-pandemic period, 2,353,170 individuals received either HCV antibody or RNA tests at least once; average age was 44.3 ± 17.8 years and 46.4% were male. During the pandemic period, 1,312,942 individuals were tested; average age was 43.0 ± 18.0 years and 45.3% were male. The distribution of demographic characteristics, including neighborhood quintile, rurality, household and dwelling quintile, material resources, as well as racialized and newcomer populations, were largely similar between the two study periods. Homelessness and HIV status were observed in up to 0.5% of people during both periods. People with a history of injection drug use comprised approximately 13.0% of the cohort, about 1.6% of individuals had advanced liver disease (including cirrhosis, decompensated cirrhosis, hepatocellular carcinoma, or had undergone liver transplantation), and the average ADG score was 6.1 during both periods (Table 1).

The characteristics of individuals with any HCV antibody test (pre-pandemic period: N = 2,337,727; pandemic period: N = 1,302,963) were practically identical to those of the overall tested cohort (Appendix A). In contrast, individuals with any RNA test (pre-pandemic period: N = 66,254; pandemic period: N = 34,560) tended to be older, more likely to be male (60% in both periods), had higher rates of a history of injection drug use (54–63%), homelessness (2.4–5.3%), and HIV (2.4–2.8%), during pre-pandemic and pandemic periods (Appendix A).

### 3.2. HCV Testing Rate

The main analysis revealed a deep and sustained disruption in overall HCV testing rates, which were increasing before the pandemic, dropped sharply and remained below expected levels throughout the three-year follow-up. The overall median monthly number of individuals who received HCV testing was 55,356 (interquartile range [IQR]: 13,512) during the pre-pandemic period and 51,857 (IQR: 8188) during the pandemic period (Appendix A). The median monthly number for antibody tests was 53,421 (IQR: 13,816) and 50,760 (IQR: 7991), and for RNA tests was 2473 (IQR: 415) and 1634 (IQR: 228) during pre-pandemic and pandemic periods, respectively.

The HCV testing rate showed a statistically significant upward trend during the pre-pandemic period (Figure 1, Appendix A). At the onset of the COVID-19 pandemic, the HCV testing rate dropped drastically, with 1.38 fewer individuals per 1000 individuals initiating testing. Following the COVID-19 period, the upward trend in the testing rate resumed, though with no significant difference in the slopes of testing rates between the pre-pandemic and pandemic periods (Figure 1, Appendix A), the testing rate remained consistently lower than the projected counterfactual estimates throughout 2020–2022.

Compared to counterfactual estimates, the monthly number of individuals tested per 1000 was lower by 1.41 (95% CI: 1.18–1.64) in 2020 (May–Dec), 1.17 (95% CI: 0.99–1.36) in 2021, and 1.41 (95% CI: 1.22–1.59) in 2022, corresponding to relative decreases in HCV testing rates of 47% (39–54%), 34% (28–39%), and 41% (35–46%), respectively (Table 2).

The HCV antibody testing rate reflected the overall trend (Table 3 and Appendix A, and Figure 2), as antibody tests comprised the majority of all HCV tests, which included either HCV antibody or RNA testing.

On the other hand, the RNA testing rates were flat before the pandemic, showing no upward trend, then declined during the pandemic and remained at a consistently lower level thereafter (Appendix A and Figure 3). Compared to counterfactual estimates, the relative decrease in monthly number of individuals with RNA tests per 1000 was 41% (33–49%) in 2020, 40% (33–46%) in 2021, and 47% (40–53%) in 2022 (Table 4).

### 3.3. Subgroup Analysis

While all subgroups experienced marked testing declines, the pandemic exacerbated pre-existing inequities among priority populations including older adults, people experiencing homelessness, PWID, and those co-infected with HIV, whereas younger individuals and immigrants showed partial recovery (Table 2, Appendix A).

Trends by sex and immigration status were consistent with those observed in the overall sample, showing a statistically significant upward trend in HCV testing rates pre-pandemic, followed by a substantial decline immediately after the pandemic, and resumed an upward trend post-pandemic (Appendix A, Appendix A).

When analyzed by birth cohort, the post-pandemic upward trend resumed for people born after 1965 but not for people born before 1965 (Appendix A, Appendix A). Compared to counterfactual estimates individuals born between 1945–1964 experienced the greatest relative declines of 62–74% during 2020–2022.

Compared to counterfactual estimates the greatest absolute decline in testing rates was observed among people co-infected with HIV, people experiencing homelessness, and PWID. During 2020–2022, the number of people tested per 1000 was lower by 8.0–10.3 among people with HIV, by 4.0–4.8 among those experiencing homelessness, and by 2.6–3.2 among PWID (Table 2). However, relative declines in testing rates were generally smaller in these priority populations compared to their non-priority counterparts, and remained small during 2020–2022, possibly due to flat or declining pre-pandemic trends that resulted in lower counterfactual estimates, rather than true recovery, as post-pandemic slopes remained flat throughout 2020–2022 (Appendix A). Among individuals with HIV, testing rates were declining pre-pandemic, then decreased and plateaued at a lower level post-pandemic (Appendix A). Among people experiencing homelessness, testing rates were relatively flat before the pandemic, then decreased and stabilized at a lower level after the pandemic (Appendix A). Among PWID testing rates showed an upward trend pre-pandemic, then decreased and plateaued post-pandemic (Appendix A).

The HCV antibody testing rates by subgroups followed a trend similar to the main analysis (Table 3 and Appendix A). The HCV RNA testing rates before the pandemic were either stable or declining across most subgroups, except among individuals born after 1975 and those experiencing homelessness, who exhibited increasing trends. At the onset of the pandemic, RNA testing rates declined sharply across all subgroups and remained consistently below projected counterfactual estimates throughout 2020–2022, except individuals co-infected with HIV. Among individuals co-infected with HIV, the post-pandemic observed monthly testing rate in 2022 was 38.5% higher than counterfactual estimates, possibly due to substantial declining trend during the pre-pandemic period rather than any recovery during the post-pandemic period. No subgroup showed an upward trend in the post-pandemic period (Table 4 and Appendix A). Figure 4 displays absolute and relative changes in testing rates in 2022 compared to pre-pandemic levels by subgroups.

## 4. Discussion

Our study examined how the pandemic and related control measures affected HCV testing rates in Ontario, particularly among priority populations. Our findings suggest that both initial screening and confirmatory testing pathways were affected. HCV antibody testing rates were increasing before the pandemic, but declined substantially at its onset. Although rates resumed an upward trend, they remained consistently below expected levels through 2022. The disproportionate impact on priority populations is particularly concerning. While younger populations and immigrants showed some recovery, older adults, people experiencing homelessness, PWID, and those living with HIV did not show similar recovery. RNA testing trends were flat or even declining across most subgroups before the pandemic, dropped further during the pandemic, and remained at a lower level thereafter.

Compared to counterfactual estimates, the greatest absolute decline in antibody testing rates was among people with HIV co-infection, experiencing homelessness, and PWID. These groups often face multiple barriers to healthcare access, likely worsened by the pandemic [34,35]. People who inject drugs were additionally affected by reduced access to harm reduction and HCV prevention services, including needle and syringe exchange programs and supervised consumption sites, and HCV testing and treatment [35,36]. The overlap between injection drug use, homelessness [26,36], and HCV–HIV co-infection [27] likely explains the similar patterns observed across these groups. Delayed diagnosis risks ongoing transmission, disease progression, and increases morbidity and healthcare costs [37]. Although these priority populations experienced the largest absolute reductions, their relative declines were smaller than those in the general population, likely not due to post-pandemic recovery, but rather because testing rates had plateaued or even declined prior to the pandemic, lowering projected levels. This plateau likely reflects both earlier and ongoing outreach efforts targeting these high-risk groups, as well as persistent social and structural barriers, such as stigma and unstable housing, which continued to limit testing despite ongoing risk.

The pandemic’s independent effect on testing volumes may be smaller than our models suggest, as other temporal trends related to HCV treatment policies and the evolving disease burden were potentially not fully controlled by our analytical adjustments. The upward trend in HCV antibody testing rates observed between 2015–2020 in Ontario might be attributable to WHO and national hepatitis elimination goals [12,38] and the increasing availability of publicly-funded direct-acting antiviral treatment in Ontario between 2015–2018, which may have driven demand for testing to assess eligibility for treatment [10]. Additionally, as increasing numbers of people have been treated and cured of HCV, the population living with chronic HCV has been decreasing [39], which may naturally reduce HCV screening needs among populations not at ongoing risk of HCV acquisition. These pre-existing trends may have contributed to testing declines independent of pandemic effects.

During the pandemic, testing rates declined dramatically, likely due to reorganization of healthcare priorities. Healthcare facilities were limiting non-urgent services to allocate resources to COVID-19 response efforts [20,40]. While screening has partially rebounded, disruptions in confirmatory testing, linkage to care, or treatment monitoring have persisted. The lack of recovery in RNA testing raises concerns about gaps in follow-up and care for those diagnosed during or after the pandemic. This trend may reflect the reorganization of services, such as the requirement for lab appointments, while disengagement during the pandemic may contribute to ongoing low testing rates. Notably, testing rates have not returned to pre-pandemic levels, with RNA testing remaining flat across all groups and no significant recovery in trends, suggesting that without targeted interventions, sustained gaps in testing are likely to persist. Understanding and addressing these barriers is key to improve access and testing uptake. Tailored interventions to reduce linkage-to-care gaps could include point-of-care RNA testing [41,42]. Outreach to highly affected populations may be improved through mobile health units and by integrating HCV screening services into shelter setting, as well as existing harm reduction and social support programs [43,44,45].

Similar declines in HCV testing have been documented in other Canadian provinces [15,46] and high-income countries such as the United States [47], England [48], and France, [49]. However, the duration and extent of recovery vary by jurisdiction. Elsewhere in Canada, a study from British Columbia reported a recovery to near pre-pandemic levels by the end of 2020 [15], while research from Alberta [46] observed a pattern similar to ours, with testing rates gradually increasing but still remaining below the pre-pandemic levels. The faster recovery in British Columbia may reflect differences in COVID-19 restrictions, which were more stringent and remained in force longer in Ontario than in British Columbia, possibly contributing to the sustained decreases in HCV testing observed in Ontario [50].

Study limitations include incomplete information in administrative data and the accuracy of the algorithms used to identify priority population. While data on HCV testing from OLIS incorporates records from laboratories across Ontario [24,25], limitations in data completeness may lead to underestimation of test volumes and introduce uncertainty in measuring temporal trends. Additionally, the accuracy of the homelessness algorithm has improved after 2018, when recording homeless status during hospital visits became mandatory [29]. Furthermore, while the high-sensitivity PWID algorithm used in this study is effective for identifying individuals who inject drugs among those with HCV, with a positive predictive value of 93%, its performance in the general population is significantly lower, with an estimated positive predictive value of only 14% [31]. This likely inflated the denominator for the PWID population, resulting in markedly lower testing rates among PWID compared to homeless individuals, despite substantial overlap in these populations. Finally, our projected testing volumes were based on the assumption that pre-pandemic trends would have continued unchanged in the absence of COVID-19. However, factors such as declining HCV prevalence may have naturally led to a stabilization or decrease in testing demand, even without the pandemic. This could result in an overestimation of the pandemic’s impact in the current analysis.

This is the second study from Ontario examining trends in HCV testing, building on prior research [17]. A major strength of our study is the focus on priority populations, identified using validated algorithms [28,51], which enhances the relevance of our findings. Additionally, we leveraged population-based data covering nearly the entire Ontario population, providing a more comprehensive assessment of testing trends that included both HCV antibody and RNA testing. Our study also employed a longer time horizon, allowing estimating both the initial impact of COVID-19 restrictions and over two years of subsequent testing data to assess recovery in HCV testing levels. The observed trends could be extended to forecast future testing rates, though current trajectories suggest a return to pre-pandemic levels is unlikely without tailored interventions. Our findings will be of value for contextualizing provincial public health surveillance HCV case report data [52] and lower rates of reported cases, in relation to changes in levels of testing.

## 5. Conclusions

In conclusion, the COVID-19 pandemic significantly disrupted HCV testing rates in Ontario, with testing levels remaining below expected projections through 2022. Challenges remain to ensure equitable access to care, underscoring the need for tailored strategies to re-engage vulnerable populations in HCV testing and care. Findings from this study will contribute to a broader understanding of pandemic-related healthcare disruptions and may inform future policy measures to mitigate service interruptions during public health emergencies.

## Figures and Tables

**Figure 1 viruses-17-01163-f001:**
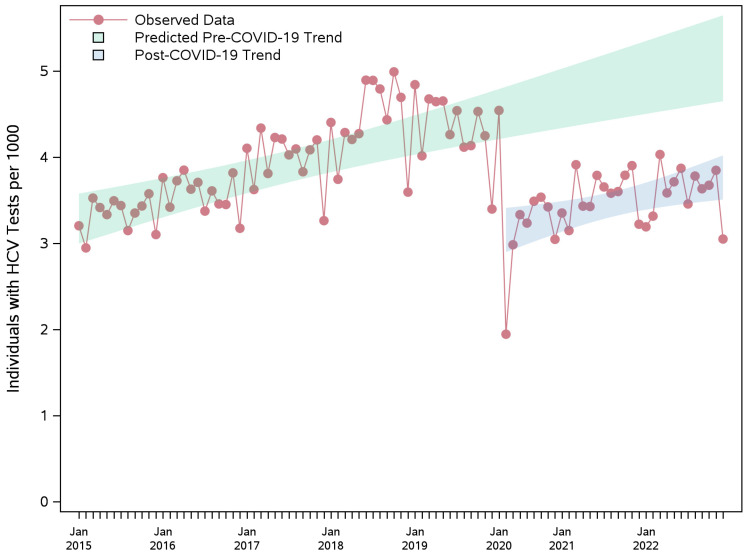
Observed monthly rate of individuals per 1000 with AB or RNA tests during pre-pandemic and pandemic periods, and the projected counterfactual pre-pandemic trend with 95% confidence band. The sharp decline in testing observed in early 2020 coincides with the onset of the COVID-19 pandemic and the provincial state of emergency, which was declared in March 2020 in Ontario. AB: antibody; HCV: hepatitis C virus; RNA: ribonucleic acid.

**Figure 2 viruses-17-01163-f002:**
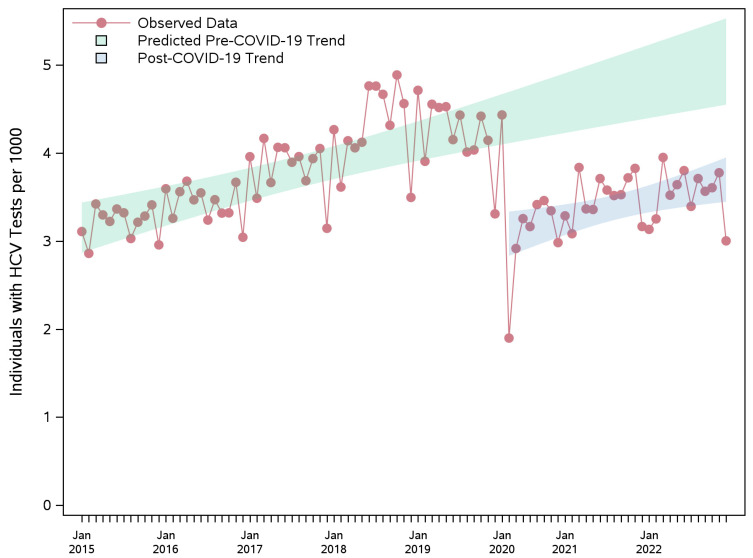
Observed monthly rate of individuals per 1000 with HCV AB tests during pre-pandemic and pandemic periods, and the projected counterfactual pre-pandemic trend with 95% confidence band. The sharp decline in testing observed in early 2020 coincides with the onset of the COVID-19 pandemic and the provincial state of emergency, which was declared in March 2020 in Ontario. AB: antibody; HCV: hepatitis C virus.

**Figure 3 viruses-17-01163-f003:**
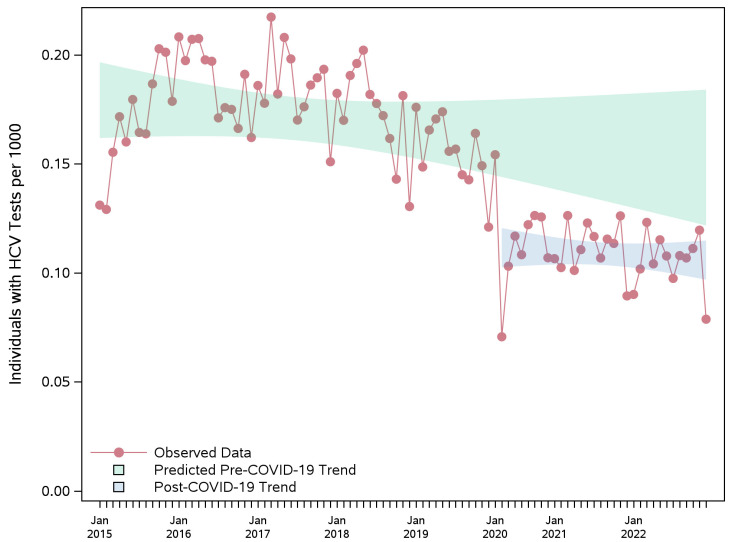
Observed monthly rate of individuals per 1000 with HCV RNA tests during pre-pandemic and pandemic periods, and the projected counterfactual pre-pandemic trend with 95% confidence band. The sharp decline in testing observed in early 2020 coincides with the onset of the COVID-19 pandemic and the provincial state of emergency, which was declared in March 2020 in Ontario. HCV: hepatitis C virus; RNA: ribonucleic acid.

**Figure 4 viruses-17-01163-f004:**
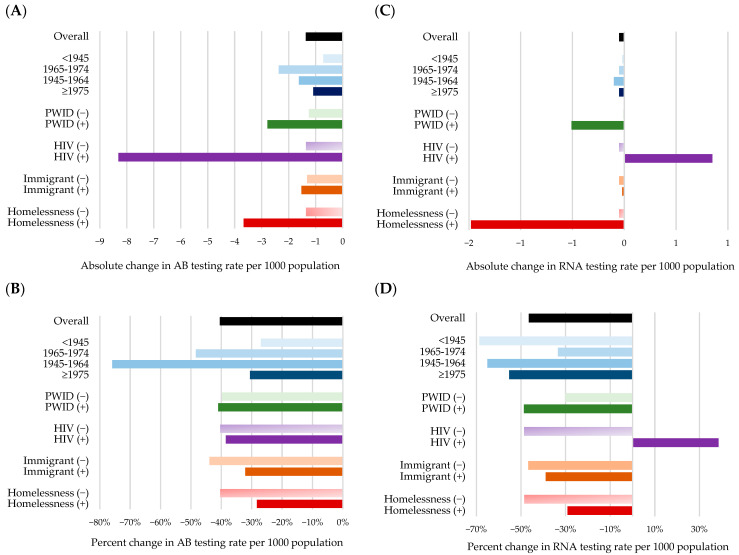
Absolute and relative changes in HCV AB (**A**,**B**) and RNA (**C**,**D**) testing rates in 2022 compared to pre-pandemic levels, overall and by priority population. AB: antibody; HCV: hepatitis C virus; HIV: human immunodeficiency virus; PWID: people who inject drug; RNA: ribonucleic acid.

**Table 1 viruses-17-01163-t001:** Characteristics of the study cohort with either antibody or RNA test at index date.

Characteristics	Pre-Pandemic 01/01/2015–29/02/2020 N = 2,353,170	Pandemic 01/03/2020–31/12/2022 N = 1,312,942
**Age at index, years, mean (sd)**	44.29 (17.75)	42.97 (17.96)
**Birth year, *n* (%)**		
<1945	156,430 (6.6)	69,307 (5.3)
1945–1965	656,973 (27.9)	293,734 (22.4)
1965–1974	380,416 (16.2)	179,677 (13.7)
>1974	1,159,351 (49.3)	770,224 (58.7)
**Male sex, *n* (%)**	1,092,990 (46.4)	595,357 (45.3)
**Immigrant status, *n* (%)**	684,444 (29.1)	385,980 (29.4)
**Rural, *n* (%)**	158,652 (6.7)	94,754 (7.2)
**Neighborhood income quintile, *n* (%)**		
1st quintile (lowest)	532,432 (22.6)	297,002 (22.6)
2nd quintile	482,521 (20.5)	270,962 (20.6)
3rd quintile	459,315 (19.5)	258,428 (19.7)
4th quintile	438,110 (18.6)	244,367 (18.6)
5th quintile	432,644 (18.4)	237,654 (18.1)
**Household and dwellings quintile, *n* (%)**		
1st quintile (lowest)	507,983 (21.6)	275,249 (21.0)
2nd quintile	391,983 (16.7)	214,853 (16.4)
3rd quintile	375,219 (15.9)	209,582 (16.0)
4th quintile	403,462 (17.1)	231,401 (17.6)
5th quintile	652,137 (27.7)	367,470 (28.0)
**Material resources quintile, *n* (%)**		
1st quintile (lowest)	501,587 (21.3)	267,262 (20.4)
2nd quintile	466,205 (19.8)	264,116 (20.1)
3rd quintile	441,675 (18.8)	253,586 (19.3)
4th quintile	435,641 (18.5)	241,679 (18.4)
5th quintile	485,676 (20.6)	271,912 (20.7)
**Racialized and newcomer populations quintile, *n* (%)**		
1st quintile (lowest)	259,613 (11.0)	147,808 (11.3)
2nd quintile	304,978 (13.0)	175,871 (13.4)
3rd quintile	381,040 (16.2)	214,499 (16.3)
4th quintile	525,753 (22.3)	294,579 (22.4)
5th quintile	859,400 (36.5)	465,798 (35.5)
**Homelessness, *n* (%)**	7028 (0.3)	6789 (0.5)
**PWID, *n* (%) (ever)**	313,189 (13.3)	173,612 (13.2)
**HIV positivity, *n* (%)**	10,959 (0.5)	7028 (0.5)
**ADG, mean (sd)**	6.12 (3.81)	6.11 (3.84)
**ADG categories, *n* (%)**		
0–3	655,147 (27.8)	373,897 (28.5)
4–8	1,110,250 (47.2)	608,403 (46.3)
9–10	270,672 (11.5)	150,390 (11.5)
>10	317,101 (13.5)	180,252 (13.7)
**Advanced liver disease, *n* (%)**		
Any advanced liver disease	38,140 (1.6)	21,438 (1.6)
Cirrhosis	36,524 (1.6)	20,440 (1.6)
Decompensated cirrhosis	8715 (0.4)	4559 (0.3)
HCC	2422 (0.1)	1170 (0.1)
Liver transplant recipients	1280 (0.1)	621 (<0.1)

This table includes anyone with an antibody or HCV RNA test, regardless of result. The characteristics are as of the index date, which is the earliest of the antibody or RNA or genotype test date, or antiviral initiation or SVR attainment in the study window. Those who tested multiple times within a given study period were counted only once. Those who were tested in both periods were counted once for each period, the same baseline characteristics applied. ADG: aggregated diagnostic group; HCC: hepatocellular carcinoma; HCV: hepatitis C virus; HIV: human immunodeficiency virus; N: number of observations; PWID: people who inject drugs; RNA: ribonucleic acid; sd: standard deviation; SVR: sustained virologic response.

**Table 2 viruses-17-01163-t002:** Observed vs. counterfactual numbers of individuals with HCV antibody or RNA tests per 1000, overall and stratified by priority groups.

Population	Absolute Difference (95% CI)			Percent Difference (95% CI)		
	May–Dec 2020	Jan–Dec 2021	Jan–Dec 2022	May–Dec 2020	Jan–Dec 2021	Jan–Dec 2022
Overall	−1.41 (−1.64; −1.18)	−1.17 (−1.36; −0.99)	−1.41 (−1.59; −1.22)	−46.5 (−53.99; −39.01)	−34.24 (−39.66; −28.82)	−40.76 (−46.13; −35.39)
Sex						
Female	−1.37 (−1.61; −1.12)	−1.13 (−1.33; −0.93)	−1.44 (−1.64; −1.24)	−41.98 (−49.51; −34.46)	−30.84 (−36.31; −25.37)	−39.69 (−45.21; −34.17)
Male	−1.46 (−1.67; −1.25)	−1.22 (−1.39; −1.04)	−1.37 (−1.55; −1.20)	−51.91 (−59.44; −44.37)	−38.23 (−43.65; −32.80)	−41.98 (−47.25; −36.71)
Birth cohort						
<1945	−0.59 (−0.78; −0.41)	−0.53 (−0.68; −0.39)	−0.74 (−0.89; −0.59)	−23.72 (−30.95; −16.49)	−19.62 (−25.03; −14.20)	−27.47 (−32.95; −21.99)
1945–1964	−1.89 (−2.12; −1.67)	−2.02 (−2.21; −1.84)	−2.42 (−2.60; −2.23)	−64.28 (−71.96; −56.61)	−62.47 (−68.17; −56.77)	−75.11 (−80.85; −69.37)
1965–1974	−1.83 (−2.09; −1.58)	−1.49 (−1.69; −1.28)	−1.70 (−1.91; −1.50)	−60.74 (−69.05; −52.43)	−42.99 (−48.89; −37.09)	−49.44 (−55.36; −43.52)
≥1975	−1.25 (−1.50; −0.99)	−0.94 (−1.15; −0.73)	−1.13 (−1.34; −0.92)	−39.69 (−47.79; −31.60)	−26.36 (−32.18; −20.53)	−31.10 (−36.84; −25.35)
IDU status (ever)						
PWID	−2.81 (−3.38; −2.24)	−2.56 (−3.03; −2.09)	−3.21 (−3.68; −2.73)	−38.07 (−45.84; −30.31)	−32.67 (−38.68; −26.65)	−42.53 (−48.78; −36.29)
Non-PWID	−1.31 (−1.51; −1.10)	−1.07 (−1.24; −0.90)	−1.26 (−1.43; −1.10)	−48.03 (−55.59; −40.46)	−34.39 (−39.79; −28.98)	−39.89 (−45.19; −34.59)
Immigration status						
Yes	−2.08 (−2.40; −1.75)	−1.47 (−1.74; −1.21)	−1.55 (−1.81; −1.29)	−50.98 (−58.97; −42.99)	−31.34 (−36.96; −25.71)	−32.27 (−37.78; −26.77)
No	−1.24 (−1.44; −1.04)	−1.10 (−1.26; −0.93)	−1.37 (−1.53; −1.20)	−45.00 (−52.33; −37.67)	−35.74 (−41.11; −30.36)	−44.32 (−49.68; −38.96)
HIV status						
HIV (+)	−9.64 (−12.49; −6.78)	−10.32 (−12.61; −8.03)	−8.00 (−10.29; −5.70)	−41.00 (−53.16; −28.83)	−47.02 (−57.45; −36.58)	−34.49 (−44.38; −24.59)
HIV (−)	−1.40 (−1.63; −1.17)	−1.16 (−1.35; −0.98)	−1.40 (−1.58; −1.21)	−46.56 (−54.07; −39.05)	−34.21 (−39.64; −28.78)	−40.82 (−46.2; −35.44)
Homelessness status						
Yes	−4.83 (−7.13; −2.53)	−3.98 (−5.83; −2.13)	−4.77 (−6.62; −2.91)	−29.82 (−44.02; −15.63)	−23.26 (−34.06; −12.45)	−28.6 (−39.71; −17.49)
No	−1.41 (−1.64; −1.18)	−1.17 (−1.35; −0.98)	−1.40 (−1.59; −1.22)	−46.65 (−54.15; −39.14)	−34.33 (−39.76; −28.90)	−40.83 (−46.21; −35.45)

CI: confidence interval; HCV: hepatitis C virus; HIV: human immunodeficiency virus; IDU: injection drug use; PWID: people who inject drugs; RNA: ribonucleic acid.

**Table 3 viruses-17-01163-t003:** Observed vs. counterfactual numbers of individuals with HCV antibody tests per 1000, overall and stratified by priority groups.

Population	Absolute Difference (95% CI)			Percent Difference (95% CI)		
	May–Dec 2020	Jan–Dec 2021	Jan–Dec 2022	May–Dec 2020	Jan–Dec 2021	Jan–Dec 2022
Overall	−1.37 (−1.59; −1.15)	−1.13 (−1.31; −0.95)	−1.37 (−1.55; −1.18)	−46.16 (−53.68; −38.65)	−33.78 (−39.20; −28.36)	−40.35 (−45.72; −34.98)
Sex						
Female	−1.33 (−1.58; −1.09)	−1.10 (−1.29; −0.90)	−1.40 (−1.60; −1.21)	−41.61 (−49.14; −34.08)	−30.43 (−35.90; −24.96)	−39.25 (−44.75; −33.74)
Male	−1.41 (−1.61; −1.20)	−1.17 (−1.34; −1.00)	−1.33 (−1.50; −1.16)	−51.73 (−59.31; −44.15)	−37.78 (−43.22; −32.35)	−41.66 (−46.93; −36.39)
Birth cohort						
<1945	−0.58 (−0.76; −0.40)	−0.52 (−0.66; −0.37)	−0.72 (−0.87; −0.57)	−23.21 (−30.51; −15.92)	−19.04 (−24.49; −13.59)	−26.99 (−32.50; −21.47)
1945–1964	−1.82 (−2.04; −1.60)	−1.97 (−2.15; −1.79)	−2.37 (−2.55; −2.19)	−64.05 (−71.83; −56.26)	−62.69 (−68.45; −56.93)	−75.89 (−81.67; −70.10)
1965–1974	−1.76 (−2.01; −1.52)	−1.42 (−1.62; −1.22)	−1.62 (−1.82; −1.42)	−60.45 (−68.82; −52.08)	−42.30 (−48.21; −36.38)	−48.38 (−54.31; −42.45)
≥1975	−1.22 (−1.48; −0.97)	−0.91 (−1.12; −0.71)	−1.09 (−1.30; −0.89)	−39.54 (−47.63; −31.45)	−25.96 (−31.77; −20.14)	−30.58 (−36.31; −24.85)
IDU status (ever)						
PWID	−2.44 (−2.95; −1.93)	−2.18 (−2.60; −1.76)	−2.79 (−3.21; −2.37)	−36.53 (−44.20; −28.86)	−30.81 (−36.74; −24.88)	−41.03 (−47.21; −34.84)
Non-PWID	−1.29 (−1.50; −1.09)	−1.06 (−1.22; −0.89)	−1.25 (−1.42; −1.09)	−47.81 (−55.41; −40.22)	−34.18 (−39.60; −28.76)	−39.79 (−45.10; −34.48)
Immigration status						
Yes	−2.05 (−2.37; −1.73)	−1.45 (−1.71; −1.19)	−1.53 (−1.79; −1.27)	−50.80 (−58.81; −42.79)	−31.13 (−36.76; −25.50)	−32.13 (−37.64; −26.63)
No	−1.20 (−1.39; −1.00)	−1.06 (−1.22; −0.90)	−1.32 (−1.48; −1.16)	−44.60 (−51.94; −37.26)	−35.24 (−40.61; −29.86)	−43.90 (−49.25; −38.55)
HIV status						
HIV (+)	−9.35 (−12.02; −6.68)	−10.31 (−12.44; −8.17)	−8.31 (−10.46; −6.17)	−43.12 (−55.43; −30.82)	−51.01 (−61.59; −40.43)	−38.51 (−48.43; −28.59)
HIV (−)	−1.36 (−1.58; −1.14)	−1.12 (−1.30; −0.94)	−1.36 (−1.54; −1.18)	−46.19 (−53.73; −38.66)	−33.72 (−39.15; −28.29)	−40.37 (−45.75; −35.00)
Homelessness status						
Yes	−3.82 (−5.65; −1.98)	−3.20 (−4.68; −1.72)	−3.67 (−5.15; −2.19)	−29.49 (−43.65; −15.32)	−23.83 (−34.81; −12.84)	−28.28 (−39.67; −16.90)
No	−1.37 (−1.59; −1.15)	−1.13 (−1.31; −0.95)	−1.36 (−1.54; −1.18)	−46.28 (−53.80; −38.76)	−33.83 (−39.26; −28.40)	−40.39 (−45.77; −35.01)

CI: confidence interval; HCV: hepatitis C virus; HIV: human immunodeficiency virus; IDU: injection drug use; PWID: people who inject drug; RNA: ribonucleic acid.

**Table 4 viruses-17-01163-t004:** Observed vs. counterfactual numbers of individuals with HCV RNA tests per 1000, overall and stratified by priority groups.

Population	Absolute Difference (95% CI)			Percent Difference (95% CI)		
	May–Dec 2020	Jan–Dec 2021	Jan–Dec 2022	May–Dec 2020	Jan–Dec 2021	Jan–Dec 2022
Overall	−0.04 (−0.05; −0.04)	−0.04 (−0.05; −0.04)	−0.05 (−0.06; −0.04)	−41.12 (−48.96; −33.28)	−39.85 (−46.33; −33.38)	−46.50 (−53.23; −39.77)
Sex						
Female	−0.04 (−0.04; −0.03)	−0.04 (−0.04; −0.03)	−0.04 (−0.05; −0.04)	−40.89 (−49.13; −32.64)	−41.12 (−47.98; −34.27)	−53.91 (−61.00; −46.83)
Male	−0.06 (−0.07; −0.04)	−0.05 (−0.06; −0.04)	−0.05 (−0.06; −0.04)	−41.51 (−49.67; −33.36)	−39.09 (−45.60; −32.59)	−41.71 (−48.56; −34.86)
Birth cohort						
<1945	−0.01 (−0.01; 0.00)	−0.02 (−0.03; −0.02)	−0.02 (−0.03; −0.02)	−9.37 (−20.50; 1.76)	−59.92 (−73.93; −45.90)	−68.67 (−83.26; −54.08)
1945–1964	−0.08 (−0.09; −0.07)	−0.07 (−0.08; −0.06)	−0.05 (−0.06; −0.04)	−49.53 (−57.63; −41.42)	−44.65 (−51.52; −37.78)	−33.50 (−40.77; −26.22)
1965–1974	−0.08 (−0.09; −0.06)	−0.08 (−0.09; −0.07)	−0.10 (−0.11; −0.09)	−51.03 (−58.99; −43.08)	−50.31 (−56.41; −44.21)	−65.09 (−71.66; −58.52)
≥1975	−0.03 (−0.04; −0.02)	−0.03 (−0.04; −0.03)	−0.05 (−0.05; −0.04)	−35.97 (−45.02; −26.93)	−37.19 (−44.05; −30.33)	−55.34 (−62.33; −48.36)
IDU status (ever)						
PWID	−0.43 (−0.53; −0.34)	−0.42 (−0.50; −0.34)	−0.51 (−0.59; −0.43)	−42.27 (−51.72; −32.82)	−38.54 (−45.75; −31.32)	−48.71 (−56.22; −41.21)
Non-PWID	−0.02 (−0.02; −0.01)	−0.02 (−0.02; −0.01)	−0.01 (−0.01; −0.01)	−39.64 (−47.97; −31.31)	−40.83 (−48.25; −33.41)	−30.13 (−37.94; −22.31)
Immigration status						
Yes	−0.04 (−0.04; −0.03)	−0.03 (−0.04; −0.03)	−0.02 (−0.03; −0.02)	−58.84 (−68.31; −49.37)	−47.65 (−55.17; −40.14)	−38.90 (−46.84; −30.97)
No	−0.05 (−0.06; −0.04)	−0.05 (−0.06; −0.04)	−0.05 (−0.06; −0.05)	−38.55 (−46.55; −30.55)	−38.44 (−44.84; −32.05)	−46.72 (−53.51; −39.93)
HIV status						
HIV (+)	−0.15 (−0.54; 0.24)	0.26 (−0.05; 0.58)	0.85 (0.54; 1.17)	−6.12 (−22.11; 9.87)	11.38 (−2.15; 24.91)	38.53 (24.32; 52.74)
HIV (-)	−0.04 (−0.05; −0.04)	−0.04 (−0.05; −0.04)	−0.05 (−0.06; −0.04)	−41.85 (−49.82; −33.88)	−41.19 (−47.55; −34.83)	−48.61 (−55.67; −41.55)
Homelessness status						
Yes	−1.31 (−1.97; −0.65)	−1.08 (−1.61; −0.56)	−1.48 (−2.00; −0.95)	−28.68 (−43.03; −14.32)	−21.42 (−31.81; −11.03)	−29.14 (−39.54; −18.75)
No	−0.04 (−0.05; −0.04)	−0.04 (−0.05; −0.04)	−0.05 (−0.05; −0.04)	−42.45 (−50.33; −34.56)	−41.90 (−48.19; −35.61)	−48.57 (−55.55; −41.59)

CI: confidence interval; HCV: hepatitis C virus; HIV: human immunodeficiency virus; IDU: injection drug use; PWID: people who inject drug; RNA: ribonucleic acid.

## Data Availability

The dataset from this study is held securely in coded form at ICES. While legal data sharing agreements between ICES and data providers (e.g., healthcare organizations and government) prohibit ICES from making the dataset publicly available, access may be granted to those who meet pre-specified criteria for confidential access, available at www.ices.on.ca/DAS (email: das@ices.on.ca). The full dataset creation plan and underlying analytic code are available from the authors upon request, understanding that the computer programs may rely upon coding templates or macros that are unique to ICES and are therefore either inaccessible or may require modification.

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
