# Peer review of "The Impact of COVID-19 and Related Public Health Measures on Hepatitis C Testing in Ontario, Canada"

_viruses, 2025, doi:10.3390/v17091163_

Round 1
Reviewer 1 Report
Comments and Suggestions for Authors
In this study the authors analysed the hepatitis C Virus clinical testing rate in Ontario, Canada, over the period 2015-2022, covering the period of disruption to healthcare provision due to the COVID-19 pandemic. The study identifies a reduction in HCV antibody and RNA testing that remains to recover to pre-pandemic levels following the 2020-2022 period. The presented data provides useful interpretation in the rates of HCV testing that will be important to revising surveillance policies to meet WHO targets. The research is not wholly novel; an earlier study by another group has identified similar reductions in HCV testing in Ontario over the period investigated. However, this study provides analysis of specific subgroups that contribute to greater understanding of the patterns of HCV testing.
I have some comments on the specific description of data in the manuscript:
Can the data presented in the manuscript be used to project the trend of future testing, predicting when testing rates might return to pre-pandemic levels? This would provide useful interpretation in the Discussion.
Table 1 and others: Birth cohort, stratified by broad year of birth is used as a parameter, without explicit justification for why this might be an important parameter for altered rates of HCV testing. Comparison of pre-pandemic and peri-pandemic groups is already analysed by age at time of sampling and is a more useful comparison. I suggest that the selection of birth cohort is justified, or the birth cohort analysis be removed from analysis.
Line 60: For the absence of doubt, it should be clarified that the screening rates presented for reference 12 related to either RNA testing or antibody testing.
Table 1: spelling mistake “Charachteristics“
Table1: provide greater resolution for the “0.0%” value presented for Liver Transplant Recipients. This value is non-zero.
Line 186: “individuals with any RNA test (pre-pandemic period: N=66,254; pandemic period: N=34,560) tended to be older, more likely to be male (60%).” Explain that both pre and peri-pandemic groups were higher. This is not explicit in the sentence.
Line 288: inappropriate sentence added in the text.
Author Response
We thank the Reviewer for their constructive feedback. Our responses are attached.

Reviewer 2 Report
Comments and Suggestions for Authors
This manuscript addresses a highly relevant and pressing public health issue: the sustained impact of the COVID-19 pandemic on HCV testing in Ontario, with a critical focus on priority populations. The study is highly timely and important. The level of English is very high, and the manuscript is well written. However, while the data presented are of substantial value, the manuscript in its current form requires minor revisions before it can be considered for publication. The main limitation lies in the Discussion section, which, while adequate, largely reiterates the results without delving into the underlying reasons behind the findings. By addressing the points below, the authors could elevate this work from a descriptive report to a high-impact paper.
Abstract
- Lines 18-19: The opening sentence is accurate but somewhat generic. Consider making it more specific to the HCV context to immediately engage readers interested in this field. For example: “The COVID-19 pandemic disrupted progress towards global HCV elimination goals by interrupting essential health services”
- Lines 33-35: Consider strengthening the conclusion by highlighting the consequences of inaction. For example: “Tailored interventions are urgently needed to address these persistent gaps in HCV testing, without which Canada’s progress toward its 2030 elimination targets is at risk.”
Introduction
- Lines 42-46: The manuscript mentions "suspension of non-essential services" and "stay-at-home orders." Consider expanding to highlight additional mechanisms. Service disruptions can be multifactorial: laboratory capacity was diverted to COVID-19 testing, community services (crucial for reaching priority populations) faced closures or reduced in-person operations. From the patient side, fear of nosocomial infection and prioritization of survival needs over chronic care likely reduced care-seeking behavior for HCV.
- Lines 49-52: While the authors correctly state that elimination efforts were underway pre-pandemic, the text may suggest that everything was on track before COVID-19. Consider clarifying that significant pre-pandemic gaps in HCV testing and care engagement persisted, especially among marginalized populations, and support this with references.
- Line 59: The term “longer-term effects” could raise questions. Is this timeframe sufficient to be considered long-term? Consider replacing with: “sustained impact beyond the initial waves of the pandemic” or “trajectory of recovery during the first X years.”
- Authors refer broadly to “HCV testing,” but one of the study’s strengths is the distinction between antibody and RNA testing. Consider inserting a sentence after line 62 indicating that disruptions may not be uniform across the HCV care cascade. The impact on initial screening may differ from that on confirmatory or follow-up testing, a distinction not fully explored beyond the initial waves of the pandemic.
- Before the aim in lines 70-72, include a clear hypothesis, particularly regarding vulnerable populations. The pandemic could result in a substantial and sustained reduction in HCV testing rates, with the most pronounced declines and slowest recovery observed among priority populations who already face significant barriers to care, such as people experiencing homelessness and PWID.
M&M
- Line 81: The pandemic period is defined as starting in March 2020. While detailed policy phases are unnecessary, it’s worth noting that this period (March 2020–December 2022) encompassed multiple phases of restrictions and re-openings. On the other hand, clarify the definition of the index date. It is defined as “earliest available HCV antibody or RNA or genotype test date or antiviral treatment initiation date.” Does this aim to capture recent diagnoses possibly tested outside of the laboratory dataset’s window?
- Section 2.2: This section contains multiple acronyms (OLIS, IRCC, ICES, RPDB, ONMARG) and algorithms. While references are appropriate, a summary table in the main text indicating variable/population, main data source, and identification method would serve as a helpful visual guide.
- Section 3, Lines 132-134: The exclusion of transitional months (Feb–Apr 2020) is justified as improving trend interpretability. However, this also removes the steepest drop in testing, potentially “smoothing” the initial shock. The authors should clarify the rationale. Although non-linear models might capture the sharp initial decline more precisely, the main goal can be to compare stable trends before and after the disruption. So, the authors do not focus on modeling the immediate impact, but rather to understand the sustained changes in testing levels beyond the initial pandemic waves. Confirm and address if this is the intended rationale.
Results
- Although logically structured, the Results section lacks narrative flow. Each subsection feels isolated. Consider adding summary sentences at the beginning or end of each subsection to highlight the main finding. For example: Section 3.2, Line 198: The main analysis revealed a deep and sustained disruption in overall HCV testing. Testing rates, which were increasing before the pandemic, dropped sharply and remained below expected levels throughout the three-year follow-up. Section 3.3, Line 252: While all subgroups experienced marked testing declines, the pandemic exacerbated pre-existing inequities. The most severe and persistent deficits were observed among priority populations, including people experiencing homelessness, PWID, and those co-infected with HIV.
- The results hinge heavily on “counterfactual estimates”, projected testing rates had pre-pandemic trends continued. This term should be clearly defined in the Methods section.
- Line 267: The authors state that “relative declines in testing rates were generally smaller in these priority populations…” Does this mean that although the absolute impact was greater in vulnerable groups, the percentage decrease was smaller compared to the general population? This could suggest either more resilient specialized services or greater “pent-up demand” in the general population. This point should be addressed.
- Figures 1-3 effectively illustrate trends. An additional figure (e.g., bar graph or forest plot) or summary table comparing key deficits by subgroup in 2022 (<1945, 1945-1964, ..., PWID, homelessness, HIV+) would help visualize who was most affected at the end of the study period.
Discussion and conclusions
- Line 267-270 (from Results): The authors note smaller relative declines in priority groups, attributing this to flat pre-pandemic trends and therefore lower counterfactuals. This is a statistical explanation. What about the clinical or public health explanation? Were services for these populations more resilient or better adapted? Or had testing rates in these groups already plateaued, limiting relative drop? Explore and elaborate in this section.
- Line 333: British Columbia reportedly recovered to near pre-pandemic testing levels by late 2020. Ontario, by contrast, remains well below expected levels two years later. The Discussion should explore reasons for this difference.
- The authors call for “tailored interventions” but should be more specific. For instance, how should linkage-to-care gaps be addressed (e.g., point-of-care RNA testing)? How can outreach to the most affected populations be intensified (e.g., mobile health units)? See, for example 10.1001/jamanetworkopen.2024.38657.
- Limitations: Lines 313-320 mention that pre-pandemic DAA treatment scale-up could have increased testing, and declining HCV prevalence may naturally reduce the need for testing. This could represent a major temporal confounder. Time series models assume the upward trend would have continued unchanged. If testing demand was already stabilizing or declining, the counterfactual may overestimate the expected number of tests, thus overestimating the pandemic’s impact. This should be acknowledged clearly in the limitations.
Minor revisions
- Line 27-29: Reword for clarity: “The observed monthly testing rate was 1.41 per 1,000 population lower…”
- Line 99-102: Two ideas are joined without proper punctuation. Please revise for clarity.
- Table 1: The word "Charachteristics" is misspelled. Also, in the footnote: the abbreviation "y: year" is not used in the table header, remove to avoid confusion.
- Line 199: Add a comma: “At the onset of the COVID-19 pandemic, the HCV testing rate dropped drastically…”
- Line 296: The sentence “…PWID and those living with HIV did not.” is incomplete. Add a verb for clarity, e.g., “PWID and those living with HIV did not show similar recovery.”
Author Response

(The authors gave the same response as above.)

Reviewer 3 Report
Comments and Suggestions for Authors
This paper investigates the long-term impact of the COVID-19 on the HCV testing in Ontario, Canada. They included more than 3 million participants in this study using the health administrative data from 2015 to 2022. And it was found that there is a sharp and sustained decline in HCV antibody and RNA test rate since early 2020. The most affected groups include people living with HIV, individuals experiencing homelessness, and people who inject drugs. This study addresses a critical knowledge gap by examining the prolonged impact of the pandemic on HCV testing.
Overall the structure of the paper is great, but it might be better to explain whether the remained low testing rates reflects disruptions in confirmatory testing, follow-up care, or broader systemic factors. Clearer interpretation would strengthen the implications for care continuity.
Minor suggestions:
- Figure 1: Would be easier to interpret the figure by highlighting the start of the COVID pandemic;
- Line 316-320: Pre-existing trend may contribute to the sharp and sustained decline is solid, especially the decreasing number of people being chronically affected. To elaborate on this aspects, probably it will be way much better to zoom in what kind of test rate is really decreasing (i.e. the following-up visit, the conformational test) given the description of the dataset, it seems that this analysis is eligible to do;
- In the discussion section, it might be worth to include the access to medical care as well or the willingness o seek medical care. i.e. Percentage of people with HCV testing vs. people have any kind of medical care record. Since it is possible that the lower testing rate is just due to the less accessibility to the medical care or less willing to go to the hospital. This may provide more insightful conclusions.
Author Response

(The authors gave the same response as above.)

Round 2
Reviewer 2 Report
Comments and Suggestions for Authors
The authors have responded excellently to all suggestions. The revisions have significantly improved the introduction, the presentation of the results, and, above all, the interpretive depth of the discussion. The manuscript has evolved from a solid data report into a high-impact scientific paper with a clear narrative and well-substantiated conclusions.